# Role of Humic Substances in the (Bio)Degradation of Synthetic Polymers under Environmental Conditions

**DOI:** 10.3390/microorganisms12102024

**Published:** 2024-10-06

**Authors:** Olga Senko, Olga Maslova, Nikolay Stepanov, Aysel Aslanli, Ilya Lyagin, Elena Efremenko

**Affiliations:** 1Faculty of Chemistry, Lomonosov Moscow State University, Lenin Hills 1/3, Moscow 119991, Russia; 2Emanuel Institute of Biochemical Physics, Russian Academy of Sciences, Kosygina Street 4, Moscow 119334, Russia

**Keywords:** microplastics, humic acids, fulvic acids, bacteria, filamentous fungi, yeasts, biodestruction, photo-oxidation, hydrolytic enzymes, soil, compost, water sources, pollutants

## Abstract

Information on the detection of the presence and potential for degradation of synthetic polymers (SPs) under various environmental conditions is of increasing interest and concern to a wide range of specialists. At this stage, there is a need to understand the relationship between the main participants in the processes of (bio)degradation of SPs in various ecosystems (reservoirs with fresh and sea water, soils, etc.), namely the polymers themselves, the cells of microorganisms (MOs) participating in their degradation, and humic substances (HSs). HSs constitute a macrocomponent of natural non-living organic matter of aquatic and soil ecosystems, formed and transformed in the processes of mineralization of bio-organic substances in environmental conditions. Analysis of the main mechanisms of their influence on each other and the effects produced that accelerate or inhibit polymer degradation can create the basis for scientifically based approaches to the most effective solution to the problem of degradation of SPs, including in the form of microplastics. This review is aimed at comparing various aspects of interactions of SPs, MOs, and HSs in laboratory experiments (in vitro) and environmental investigations (in situ) aimed at the biodegradation of polymers, as well as pollutants (antibiotics and pesticides) that they absorb. Comparative calculations of the degradation velocity of different SPs in different environments are presented. A special place in the analysis is given to the elemental chemical composition of HSs, which are most successfully involved in the biodegradation of SPs. In addition, the role of photo-oxidation and photoaging of polymers under the influence of the ultraviolet spectrum of solar radiation under environmental conditions on the (bio)degradation of SPs in the presence of HSs is discussed.

## 1. Introduction

The already accumulated and projected volumes of waste of difficult-to-decompose synthetic polymers (SPs) in the form of fabrics, plastics, packaging materials, equipment parts, etc., represent a global problem for the environment [1,2]. It is known that the decomposition of polymers in environmental conditions is accompanied by the formation of hazardous microplastics (MPs) with particle sizes from 1 µm to 5 mm [3]. The problem of environmental pollution with MPs is being addressed today in several directions simultaneously: 1—the development and application of environmentally friendly biodegradable polymers, which are intended for partial replacement of traditional SPs [4,5]; 2—the development of processes for physicochemical processing and reutilization of SPs [2,6]; 3—the search for new opportunities for deep degradation of MPs. The problem solution supposes the use of biocatalytic systems, including consortia of microorganisms (MOs) and their or additionally introduced enzymes [7,8].

The rate of MP decomposition under natural conditions is low, which leads to their gradual accumulation in soil and water. The small size of MPs can easily penetrate into the tissues and organs of living organisms, which leads to the generation of reactive oxygen species (ROS), the manifestation of local toxicity, mutagenesis [9], and inhibition of metabolic processes in cells, including those associated with photosynthesis [10]. MPs accumulating in cells are transmitted through food chains [11], negatively affecting participants in the processes of substance circulation in nature, changing the composition of biocenoses, primarily microbial communities, and threatening the sustainable existence of ecosystems. An additional negative effect from the presence of MPs is recorded when various carcinogenic additives (flame retardants, UV stabilizers, colorants, etc. [3]) are used for SP production and present in the content of MPs. In natural conditions, MPs also easily absorb and retain various compounds (antibiotics, pesticides, viral proteins, mycotoxins, etc.), which introduces an additional factor of their possible negative impact on living objects.

Today, various methods of plastics processing are used to treat large volumes of SP waste, whereas it seems more appropriate to use biocatalytic approaches [12] or hybrid processes combining physicochemical processing and biocatalysis to overcome problems with MPs, most of which are already in the environment (in water and soil systems) [13,14]. Researchers in experimental studies highlight the role of humic substances (HSs), present in the composition of soils and natural waters, in the processes associated with the degradation of SPs [15,16,17,18]. However, a general systematization of the information accumulated to date on this issue seems to be extremely relevant and has important scientific and practical significance, allowing us to draw a conclusion about the possibility of using HSs to regulate the processes of (bio)degradation of SPs in situ.

HSs are complex heterogeneous polydisperse nitrogen-containing compounds of phenolic nature, formed as a result of the decomposition of plant and animal remains under the influence of environmental factors and MOs [19]. In the composition of HSs, fulvic and humic acids (FA and HA, correspondently) are distinguished, certain ratios of which determine the presence of different functional (alcohol, phenol, aldehyde, carbonyl, ketone, quinone, peptide, nitrogen-containing) groups in the structures of HSs and, accordingly, the properties of these substances [20]. The nature of the impact of HSs on the metabolism of MOs is determined by their characteristics and concentrations present [21], which can suppress or activate metabolism in microbial aerobic and anaerobic biosystems [22]. In addition, under environmental conditions, HSs are adsorbed on the surface of SPs and, under the influence of sunlight ultraviolet (UV) rays, cause the formation of ROS, which intensify the process of abiotic photodegradation of MPs in aquatic and soil ecosystems [23]. Such phenomena promote the (bio)degradation of various pollutants adsorbed on SP particles [19]. Some HSs can activate not just single cultures but several members of indigenous microbial consortia and, thus, accelerate biocatalytic degradation of polymers in the environment [24].

To date, there is still no analysis of the influence of different HSs on the (bio)degradation of SPs, including MPs. This is so as researchers often simply detect the presence and accumulation of polymer particles in ecosystems but do not analyze the presence of certain concentrations of HSs and MOs in them, or they determine the presence and change in the composition of bacterial communities along with polymers but do not control the presence of HSs in any way. Other researchers have traditionally studied the effects of different HSs on microbial community composition and plant growth without analyzing the presence of MPs.

In this regard, the objective of this review was to critically analyze current information on the impact of HSs on the processes of the (bio)degradation of SPs, including MPs, as well as on the most favorable characteristics of HSs and the conditions for their use to solve the discussed environmental problems and predict the prospects for methods of further application of natural and artificially obtained HSs in the processes of the (bio)degradation of polymers. The main objects of study in the review were SPs, HSs, and MOs and various aspects of their mutual influence under environmental conditions (Figure 1).

Since, ultimately, the greatest interest is presented by the processes involving SPs, HSs, and MOs, occurring precisely under environmental conditions, special attention in the review was given to the possible additional influence of the photo-oxidation of HSs and photodegradation of SPs under the influence of the UV spectrum of solar radiation on the functional activity of MOs in the discussed processes of polymer biodegradation. Papers published within the past 10 years focusing on the biodegradation of SPs in the presence of HSs were taken into account mainly to analyze current trends in their developments. The articles were searched in several scientific databases (https://scholar.google.com, https://pubmed.ncbi.nlm.nih.gov—the last accessed 2 September 2024) by using combinations of the following keywords: synthetic polymer, biodegradation, humic, substances, humic and fulvic acids, photo-oxidation, compost, etc.

## 2. Effect of the Presence of HSs on the Biodegradation of SPs

### 2.1. Analysis of Theoretical Capabilities of MOs in the Degradation Processes of SPs in the Presence of HSs

It is known, when referring to the literature, that various MOs are capable of participating in the biodegradation of SPs. Among the MOs, there are representatives of bacteria, filamentous fungi, yeast, and phototrophs. HSs are always present in various natural aquatic and soil ecosystems. It was initially interesting to conduct a theoretical analysis of the possible interactions of the three main participants in the processes under discussion (SPs, HSs, and MOs) during polymer degradation (Figure 2, Table 1 [25,26,27,28,29,30,31,32,33,34,35,36,37,38,39,40,41,42,43,44,45,46,47,48,49,50,51,52,53,54,55,56,57,58,59,60,61,62,63,64,65]). For this analysis, the information sources in which the biodegradation of different SPs under the influence of certain MOs were studied and selected in one block. The information sources in which other authors studied the effect of different HSs on the metabolism of precisely those MOs that are capable of carrying out the biodegradation of SPs were selected in the second block (Table 1). Comparison of these two information flows allowed us to theoretically evaluate the potential influence of HSs on the ability of MOs to carry out the destruction of SPs (Figure 2). In parallel, the identification of those MOs for which the presence of HSs can potentially have a positive effect on the results of the biodegradation of SPs was carried out.

Analysis of the data in Table 1 allowed us to conclude that among bacteria capable of degrading various polymers were representatives of the genera *Streptomyces* [25,26], *Bacillus* [27,28], *Pseudomonas* [29,30,31], *Rhodococcus* [32,33], *Klebsiella* [34,35], *Micrococcus* [36,37]. As the analysis showed, among various SPs, bacteria are most effective in biodegrading polyethylene terephthalate [25], polysterene [27], and poly (vinyl chloride) [29].

It is difficult to biodegrade ethylene vinyl acetate [34] and polystyrene [31] with cells of the genus *Pseudomonas*. The efficiency of polyethylene biodegradation varies greatly depending on the type of bacteria and can range from 2% [35] to 19% [28]. Some bacteria can use HSs as the main carbon source. In particular, cells of the genus *Streptomyces* are capable of breaking down high-molecular-weight HSs into aliphatic and aromatic compounds and converting them into their derivatives with a lower molecular weight. Such breakdown may be associated with the manifestation of laccase activity by cells [54].

A positive effect of HSs on the growth of 108 strains of soil bacteria of the genera *Bacillus*, *Pseudomonas*, *Actinomyces*, and *Mycobacteriumza* was established due to their possible use of HSs not only as sources of carbon and nitrogen [55]. The addition of HSs to nutrient media for bacteria of the genus *Pseudomonas* increased the concentration of cells in vitro in media with SPs [56].

The stimulating effect of HSs on bacterial growth and metabolic activity was most clearly observed in the stationary phase of MO growth, which is usually characterized by nutrient deficiency and accumulation of metabolites (Figure 3). Bacterial cells adsorb HSs on their surface, with HA adsorption greater than FA. HA affects the permeability of the outer membrane of cells and also increases the ability of bacteria to absorb nutrients from the environment.

There is evidence that HSs protect MOs from the effects of toxic compounds [56]. A stimulating effect of HA isolated from peats of various origins at a concentration of up to 50 mg/L on the growth and reproduction of *Rhodococcus* cells has been revealed. It has been shown that low-molecular HSs are capable of penetrating the cell membrane of bacteria [57]. The biological effects observed as a result of contact between HSs and bacteria were associated with the ability of cells to more effectively use the energy they produce and store in the form of ATP for the regeneration of cell components, growth, and reproduction.

In addition, HSs can act as adaptogens of cells to the presence of various eco-pollutants by stimulating the formation of a multimolecular layer of polysaccharide and amino acid residues on the surface of bacteria [57]. The greatest increase in cell biomass when using HSs as the sole source of carbon and energy was found in bacteria of the genera *Pseudomonas* and *Micrococcus* at a concentration of 0.05% and 0.125% HSs in the medium, respectively [59].

HSs have also been shown to be involved in extracellular electron transfer. For example, FA caused an increase in the number of electroactive bacteria of the genera *Bacillus*, *Desulfosporosinus*, *Klebsiella*. *Klebsiella* cells exhibited electrogenic effects by secreting quinone compounds as electron transfer mediators. Interspecies electron exchange proved to be vital for biostimulating degradation of organic pollutants, especially in soils where mass transfer is difficult [58].

Each of the above effects of HSs on MOs or their combination can be the cause of an increase in the number of bacteria in the presence of HSs. However, using the example of four methanogenic consortia, it was shown that the presence of HA in high concentrations (5–10 g/L) in the biosystem leads to inhibition of the metabolic activity of cells, while FA, on the contrary, in some consortia contribute to the intensification of biogenic processes, in particular, methanogenesis [21].

Among the filamentous fungi capable of degrading SPs, producers of the genera *Aspergillus* [38,39,40], *Penicillium* [41,42], *Fusarium* [43], *Alternaria* [44] stand out. The biodegradation of SPs by the fungi is carried out under the action of various hydrolytic enzymes secreted by them (cutinases, lipases, proteases, carboxylesterases, etc.) [66]. As a result, for example, poly(3-hydroxybutyrate) and poly(3-hydroxybutyrate-co-3-hydroxyvalerate) were biodegraded by 99%, and poly (vinyl alcohol) by 81% [41,42]. Poly (vinyl chloride) [39], polystyrene [40], polyurethane were the least biodegradable under the action of *Alternaria alternata* fungus cells [44] and low-density polyethylene under the action of *Fusarium oxysporum* [43].

Unlike bacteria, suppression of the metabolic activity of fungi in the presence of HSs has been demonstrated. One of the probable mechanisms of inhibition of fungal metabolism is a non-covalent binding of hydrolytic enzyme molecules to HSs and their inactivation, leading to a decrease in the number of available substrates entering the cells [67].

Some HSs have been shown to have fungicidal activity against filamentous fungi of the genera *Fusarium* and *Sclerotinia* [62,68]. Ultrafine sapropel suspensions have an inhibitory effect on *Aspergillus niger* cells, which rises with increasing HS concentration [60].

The presence of 10 g/L HSs in the medium causes a decrease in the growth of the fungi *Penicillium digitatum* [61]. This effect is attributed to the presence of heavy metal ions, particularly copper, in HSs. Similarly, a fungicide based on HA from lignite and Cu (II) showed strong antifungal properties against *Rhizoctonia solani* [69].

It is known that quinones, having electron-acceptor properties, play an important role in the formation of ROS and, as a consequence, cause oxidative stress in fungal cells. It was also shown that HSs do not have an inhibitory effect on *A. alternata* cells at a concentration of 0.2 g/L but, on the contrary, stimulate the accumulation of biomass [63]. The high-stress resistance of *A. alternate* is associated with the ability of these cells to synthesize an increased content of melanin in response to the action of HSs, which contributes to the neutralization of ROS.

Yeasts capable of degrading SPs belong to the genera *Vanrija* [45], *Rhodotorula* [46], *Meyerozyma* [47], *Candida* [48]. The highest degradation efficiency by yeast among a number of SPs was observed for polyethylene terephthalate (10%) [45] and polyethylene (13.9%) by *Meyerozyma guilliermondii* cells [47]. The degradation efficiency of polyethylene was 3.2–3.8% [46,48].

It has been found that HSs can inhibit yeast growth because they can form a layer on the cell surface that hinders substrate access to the cells [64]. In *Candida albicans* cells exposed to HSs at a concentration of 40–80 mg/L, an increase in catalase and superoxide dismutase activity by more than two times compared to the control was observed. This increase in enzymatic activity is associated with the induction of ROS generation caused by HA, which led to lipid peroxidation and cell damage [70]. However, at low HS concentrations (10–20 mg/L), an increase in yeast biomass was observed.

Thus, in the case of filamentous fungi and yeasts, the effect of HSs on the metabolic activity of cells primarily depends on the concentration of HSs in the medium, as in the case of bacteria. However, the concentrations that have a negative effect are lower for fungi than for bacteria.

The biodegradation of SPs by a number of phototrophic MOs has been demonstrated, including representatives of the genera *Anabaena*, *Scenedesmus* [49], *Chlorella* [50], *Picochlorum* [51], *Oscillatoria* [52], *Nostoc* [53]. The cyanobacteria *Anabaena spiroides* and the microalgae *Scenedesmus dimorphus* degraded low-density polyethylene with an efficiency of 8.2 and 3.7%, respectively [49].

Polyethylene terephthalate weight loss was 5.5% after 30 days of exposure to *Chlorella vulgaris* cells [50]. The highest percentage of degradation of low-density polyethylene was found as a result of exposure to cyanobacterial cells *Oscillatoria subbrevis* (30%) [52], *Nostoccarneum* (27%) [53], and microalgae *Picochlorum maculatum* (20%) [51]. It was found that the surface of particles of low-density polyethylene is more easily colonized by microalgae and cyanobacteria than other SPs [71].

The biodegradation of SPs by cells of phototrophic MOs occurs as a result of the colonization of SPs particles due to the secretion of extracellular exopolysaccharides (EPSs) by cells and subsequent enzymatic oxidation of polymers under the action of secreted microbial enzymes (laccase and manganese peroxidase) [52].

There are relatively few studies related to the effects of HSs on microalgae. However, HSs can act as biostimulants for the growth of phototrophic MOs and the formation of cellular tolerance to various toxic substances [65]. HSs can improve the availability of mineral nutrients for phototrophs, increasing the growth rate of microalgae and their accumulation of biomass.

Low concentrations of HSs stimulated phototrophic growth and photosynthetic efficiency, while HA concentrations above 0.003% already had an adverse effect on *Chlorella vulgaris* cells. The rate of photosynthesis decreased, and respiration increased sharply. The prokaryotes are generally more sensitive to HSs than eukaryotic cells [65]. Like bacteria and fungi, microalgae use HSs as sources of C and N, but only at low concentrations in the environment. At higher concentrations, the effects of HSs on microalgae are determined by their interference with the functioning of the cellular photosynthetic apparatus and the production of pathophysiological levels of ROS.

Discussing the interaction of HSs and MOs, it should be noted that many cells participate in the processes of cell adhesion/immobilization on flocculi formed with the participation of HSs [12]. In this case, the possibilities for the formation of highly concentrated cell populations and the manifestation of quorum sensing by them are realized. As a result, the efficiency of biocatalytic processes with the participation of such cells increases, acquiring increased resistance to the negative impact of various factors in the quorum state [72]. The binding of such HSs to SP particles promotes the formation of biofilms on the surface of polymers [16].

Thus, comparison and analysis of disparate information sources (Table 1) showed that, theoretically, at certain low concentrations, HSs can stimulate the growth and metabolism of cells of different MOs capable of biodegrading SPs. The general hypothetical list of such MOs included representatives of pro- and eukaryotes, and the list of polymer destructors itself turned out to be quite large and representative.

It was of further interest to evaluate which of the theoretically identified possible combinations of MOs and HSs had actually already been investigated in in vitro and in situ conducted experiments for the degradation of SPs. This was the subject of the next stage of the analysis made in this review. Primary attention was paid to the location and conditions of polymer biodegradation, the source of HSs involved in the process, and thus, determining the properties of the HSs used, as well as the type of biocatalyst, i.e., MOs.

### 2.2. Effect of HSs on the Biodegradation of SPs In Vitro

As shown by the analysis of a very small number of recent publications identified, laboratory studies have used MOs that are common in the environment. These in vitro studies, which simulated the biodegradation of SPs in aqueous systems containing, as a rule, low concentrations of HSs (Table 2) [73,74,75,76], were primarily aimed at understanding the basic patterns of interaction between all participants in the process rather than achieving maximum efficiency of SPs degradation.

It was shown that the biodegradation efficiency of MPs of polypropylene under the influence of biofilms was significantly enhanced compared to the control, which had only the addition of 5 mg/L HA, and after 36 days, the polymer mass loss in the presence of both MOs and HSs was 12.3% [73]. The enhancement of SP biodegradation was associated with an increase in the number of MOs in the biofilms in the presence of HA due to the modification of the polypropylene surface, which contributed to better adhesion of MOs to the polymer particles. Due to the fact that HSs exhibit the properties of surfactants, they modify the surfaces of MP particles, thereby improving their accessibility to the action of enzymes responsible for the biodegradation of SPs.

Another study revealed another positive effect of HSs on the biodegradation process of MPs, the surface of which contains pollutants, in particular, pesticides widely used in agriculture [74]. Such pollutants are found in the environment in concentrations from 0.1 to 1000 μg/L as part of the complexes they form with MPs. The biodegradation of amino-functionalized polystyrene nanoplastics (0.05–0.4 mg/L) containing such a herbicide as atrazine (10 μg/L) under the action of *C. vulgaris* microalgae cells in the absence and presence of HA (1 mg/L) was studied [74] (Table 2). It has been shown that the addition of HA significantly reduces the toxicity of the pesticide–MP complex to *C. vulgaris* cells by changing the ζ-potential of its surface, creating a kind of “layer” between the cells and the toxic surface of the polymer, thus reducing the level of ROS response generation in the cells while simultaneously improving cell contact with the polymer surface.

The microalgae *C. vulgaris* themselves were capable of secreting exopolysaccharides, the presence of which accelerated cell adhesion to polystyrene and polymer degradation due to oxidative processes and the formation of hydroxyl groups on the surface of SPs [75]. As a result of polystyrene degradation, compounds similar to HA are formed, which replenish the amount of dissolved organic matter. Hybrid modification of hydroxypropyl methyl cellulose with various types of HA made it possible to slow down but maintain the biodegradability of polymer films due to HA concentrations that inhibit the metabolism of mycelial fungi and to achieve 91% polymer degradation in six months [76].

Thus, in vitro studies have confirmed the theoretically expected direct relationship between the presence and concentrations of HSs in media with SPs and the modification of the latter. The availability of HSs leads to improving their biodegradation, enhancing microbial respiration, and increasing the concentration of the biomass of active biofilms that carry out the biodegradation of SPs, including MPs, starting from the surface layers of polymeric materials.

### 2.3. Effect of HSs on the Biodegradation of SPs In Situ

#### 2.3.1. Biodegradation of SPs in Water Ecosystems Containing HSs

It has been established that HSs play a significant role in the distribution of MPs in natural aquatic environments [77]. HSs successfully interact with MP particles, forming aggregates that sediment to the bottom of water bodies. The results showed that the average content of MPs in the bottom sediment layer (up to 10 cm) was significantly lower than in deeper layers (11–20 cm and 21–30 cm). In this case, larger MP particles were present in the upper layer (up to 10 cm), while small particles were concentrated in deeper layers (21–30 cm). Such an analysis seems important for understanding which MOs can potentially participate in the biodegradation of SPs that have such a distribution and localization in bottom sediments.

Studies of xenobiotic distribution in the water column and in bottom sediments have shown that HSs and MPs bind various pollutants, thus facilitating their distribution in aquatic ecosystems [78]. Serious problems for the biodegradation of MP particles are associated with the adsorption of pollutants such as antibiotics, the increasing concentrations of which are found in the environment due to their low-efficiency degradation at treatment facilities [79]. Both SPs (including MPs) and HSs are capable of adsorbing and binding a fairly wide range of xenobiotics [80,81,82,83].

Today, the first results of the study of biodegradation of SPs in the presence of HSs in aquatic (fresh and marine) natural systems are available (Table 3 [24,84,85]). It was found that the sorption activity of SPs with respect to various pollutants increases with aging, which often leads to the formation of non-covalent complexes that have higher toxicity than the original SPs and xenobiotics taken separately. In particular, the effect of HA on the adsorption of the antibiotic norfloxacin (NOR) in an aqueous medium on polyethylene terephthalate, polylactide, and polypropylene particles treated with a strong oxidizer (K_2_S_2_O_8_) was studied [85]. It turned out that the adsorption capacity of polylactide, polyethylene terephthalate, and polypropylene decreased with an increase in the concentration of HA (0–8 mg/L), which was caused by the competition between HA and SPs in their adsorption interaction with the antibiotic.

It has been shown that HA can interact with hydrophilic and polar organic substances in the aquatic environment through hydrogen bonding and complexation [24]. In this regard, HA can simultaneously reduce the likelihood of toxic complex formation as a result of the interaction of SPs and ecotoxicants, which ultimately provides the opportunity for MO cells to come into contact and more effectively degrade both of these pollutants (SPs and xenobiotic). In addition, it turned out that HSs provide MO cells with protection from MP particles and the toxic effects of xenobiotics (in particular, this was shown using the example of Cl-containing MP decomposition products that are toxic to cells). This effect of HSs generally contributes to the intensification of SP biodegradation [86].

The results of the conducted studies confirmed the role of HSs in the possible stimulation of the metabolic activity of MO biodegraders of SPs due to the induction of the synthesis of enzymes necessary for the biodegradation of polymers by cells. Using labeled 13C in the composition of polyethylene, the rate of its biodegradation in various types of freshwater was determined. It turned out that the rate of biodegradation in the waters of the lake containing HSs was almost 6.5 and 22 times higher per year (Table 3) than in clean lake waters and an artificially prepared freshwater environment, respectively. The main bacteria involved in the biodegradation of SPs in this study were representatives of the *Acetobacteraceae* and *Comamonadaceae* families. In addition, it turned out that these MOs in lakes whose waters contain HA-degraded polyethylene more effectively than MOs in other water bodies with a lower concentration of HSs [84].

The efficiency of polycaprolactone degradation in laboratory seawater reached 20–100% within 6–12 months with a plastic sheet thickness of 0.2–2 mm. In real seawater, a higher (2–2.5 times) degradation rate was noted for polycaprolactone film [24]. The rate of thickness loss of samples in the sea was 89 μm per month, i.e., it was almost three times higher than in the laboratory aquarium (30 μm per month). The degradation rate of plastic was primarily influenced by the temperature of the water environment, the presence of mechanical effects (sea waves, currents), as well as the presence of MOs and dissolved organic matter composed of HSs. The mass loss of polybutylene succinate and polybutylene adipate terephthalate in seawater was more than 30%.

It should be noted that the use of conditionally biodegradable SPs in the event of their release into natural water bodies may lead to difficult-to-predict consequences since the appearance of biodegradable polymers leads to changes in the composition of HSs in water bodies. It has been shown that polylactide causes changes in the molecular structure of HA, which are primarily associated with a change in the number of intermolecular hydrogen bonds and hydrophobic interactions. Carboxyl groups of HA are involved in the formation of dimers upon contact with polylactide–MPs [16]. According to the results of this study, the number of HSs adsorbed by biodegradable polylactide is relatively small compared to non-biodegradable MPs.

In general, in the course of studies of the degradation of polyethylene, polycaprolactone, polylactide, polyethylene terephthalate and polypropylene in situ in aqueous environments, it was found that HSs bind to MPs particles (i) due to the formation of multiple hydrogen bonds and hydrophobic interactions, thus promoting their sedimentation on the bottom of water bodies and protecting cells from toxic semi-products of plastic degradation; (ii) due to competitive adsorption interactions with hydrophilic and polar groups of organic substances, preventing the formation of toxic polymer complexes with xenobiotics (antibiotics) and improving the availability for biodegradation of both types of pollutants (plastics and xenobiotics), stimulating the biodegradation activity of MO destructors.

#### 2.3.2. Biodegradation of SPs in Soil Containing HSs

At present, the processes of biodegradation of SPs in soil are studied more actively than in water (Table 4 [17,24,87,88,89]), since in the case of aquatic systems, SPs are concentrated mainly not in the aquatic environment, but at the bottom, whereas in the ground they are clearly concentrated, retained for a long time, affecting agricultural production, including due to natural precipitation and the use of irrigation water that may contain MPs.

In the study of biodegradation of such SPs as polylactide, poly(3-hydroxybutyrate), polybutylene adipate terephthalate, and polyethylene in soil, an increase in the HS content over the course of a year was confirmed (especially for slowly degrading polyesters, polylactide, and polybutylene adipate terephthalate). These changes were accompanied by the decomposition and mineralization of SPs, which reached almost 25% for polybutylene adipate terephthalate and polylactide, more than 90% for poly (3-hydroxybutyrate), and 15% for polyethylene. The proportion of carbon transferred from SPs to the organic pool of the soil was very low for all the polymers studied [88]. A significant change in the concentration of FA was found during the biodegradation of poly(3-hydroxybutyrate). The authors of the study associated this with an increase in the biomass of fungi that used FA as an additional source of carbon nutrition.

A stimulating effect of HA on the growth of bacterial biofilms on the surface of polyethylene in a soil environment was established. The maximum effect (increasing roughness of the E surface of low-density polyethylene by 2.0–3.5 times) was achieved within 14 days with HA concentrations in the range of 50–250 mg/L. The positive effect of HA was associated with increased bacterial adhesion to the polymer surface and with the induction of the synthesis of bacterial enzymes corresponding to the increasing amount and type of available nutrients [87].

The degradation enhancement mechanism of dibutyl phthalate, a plasticizer used in the formation of poly (vinyl chloride) products, by HA in soil was analyzed using Fourier-transform infrared spectroscopy. It was shown that two major functional groups (C (carbonyl) = O from alkyl esters and C (aryl)–O) were found at the polymer binding site with HA. In addition, HA effectively stabilized the composition of the soil microbial community and promoted the growth of aerobic bacteria, which accelerated the degradation process of SPs [89]. The half-life of dibutyl phthalate was effectively reduced by 3.5 times (from 11.65 to 3.36 days) after the addition of HA to soil.

It should be noted that such results make it quite logical to conclude that it is possible to purposefully introduce HS sources into the soil as a matrix carrier for the MOs degrading SPs.

Polyacrylamide degradation (Table 4) by 69% was achieved within 30 days at pH 6.6 and 38 °C after introducing *Klebsiella* sp. cells immobilized on biochar (70 mg/g soil) obtained by pyrolysis of sewage sludge and coconut shell into the soil. Polyacrylamide was successfully hydrolyzed by bacterial amidase to ammonia, which was then oxidized by soil-nitrifying bacteria. The presence of biochar with introduced SP degraders contributed to the local accumulation of MO biomass, which carries out polyacrylamide biodegradation [17].

The mass loss of polycaprolactone, polybutylene succinate, polybutylene adipate terephthalate in the soil over 6 months was 2%, 1%, and 0.2%, respectively. The addition of vermicompost to the soil (in a ratio of 7:3), containing HSs, nitrogen, and phosphates in a bioavailable form, activated the growth of soil MOs, which led to a 100% degradation of polycaprolactone and polybutylene succinate [24].

Thus, in the soil, there is a close relationship between the processes of MO growth and the accumulation of their biomass, accompanied by the decomposition of SPs. Using the examples of polyacrylamide, polybutylene adipate terephthalate, polybutylene succinate, polycaprolactone, poly (3-hydroxybutyrate), polylactide, dibutyl phthalate, polyethylene, low-density polyethylene, it was found that additional introduction of HSs into the soil contributes to an increase in the amount of available nutrients, increased bacterial adhesion to the surface of polymers, the growth of destructor bacteria, stimulation of their enzymatic activity, and stabilization of the microbial community in the soil ecosystem.

#### 2.3.3. Biodegradation of SPs in Composting

Composting of plastic waste is a well-known process, considered an ecologically friendly action (Table 5 [18,90,91,92,93,94,95,96]). During composting, HSs are actively formed. The content of HSs changes during composting and, as a rule, lies in the range of 25–150 g/kg [20].

Depending on the composition of SPs, their composting efficiency varies greatly. For example, when composting modified polypropylene films after abiotic treatment (accelerated weathering for 40 h), the mass loss of samples of different compositions after 45 days was 11.2–36.4% [91].

Under normal composting conditions, the degree of decomposition of acrylic acid-grafted polypropylene samples did not exceed 5.6% over 45 days [93].

The degradation of three different materials (polycaprolactone and starch mixture (60/40 *w*/*w*), polyethylene and polyester) during controlled composting in the laboratory for 50 days was 88%, 1.1%, and 60%, respectively [92]. It was found that polyethylene is not biodegradable according to standard biodegradation tests, but in real soil, under the influence of heat and sun, the material fragments and then biodegrades. In another study, it was shown that even with an increase in the composting period to 125 days, polyethylene is practically not biodegradable (0.25%) [94].

Co-polyester poly(succinate-co-glutarate-co-adipate-co-terephthalate 1,4-butylene) in the form of a non-woven material (10 × 10 cm) was subjected to composting with the addition of plant biomass (8 and 16% of the dry weight of the total composted mass) for 90 days at different temperature conditions (25–30 °C and 40–45 °C) [95]. As a result of composting at low temperatures, material residues (1 × 1 cm) were found. In the range of 40–45 °C with 8% compost loading with plant biomass, single fibers of about 1 cm in length were found. Whereas with 16% loading, the formation of numerous fibers was observed [95]. Apparently, the increased concentration of the introduced lignocellulosic raw materials (plant polymers) was converted into HSs and slowed down the process of biodegradation of SPs.

Of interest were the degradation results of fibers made from polylactide or a mixture of polylactide with polyhydroxyalkanoates, as well as a bicomponent fiber (BICO) produced from polybutylene succinate and polylactide obtained during composting of the indicated SPs for 4 weeks at 58 °C [18]. All samples of fibrous materials were characterized by a decrease in mechanical strength and molecular weight.

The biodegradability of polycaprolactone particles (180.7 μm) during controlled composting carried out at 58 °C was 79.9% in 56 days [96].

Unfortunately, not all authors of the publications analyzed in this review controlled the concentrations of HSs in the reaction media of the processes carried out since they did not fully realize the importance of this parameter and the role of HSs in the studied biodegradation of SPs. This fact once again emphasizes the relevance of the issue of the relationship between SPs, MOs, and HSs, which is raised in this review.

Thus, in the course of in situ studies of degradation of different SPs (polypropylene, polycaprolactone, polyethylene, and mixed polymers (polylactide polybutylene succinate, polylactide–polyhydroxyalkanoates, etc.)) with the addition of different pro-oxidant loadings to the polymers or without them under composting conditions accompanied by humification of the reaction media components, it was established that, in general, the process is characterized by high efficiency only with its long duration (more than 45 days). The efficiency of SP destruction depends on the chemical composition of the polymers, as well as the temperature of the process. However, all experiments confirmed the fact that composting today provides an incomplete decomposition of SPs; the presence of MPs is recorded as a result of the process, which is unacceptable according to regulatory indicators [97].

#### 2.3.4. Biodegradation of SPs in Anaerobic Digestion

It is known that methanogenic consortia are specifically used under controlled conditions at treatment facilities to purify wastewater from various pollutants, and they also actively function independently under environmental conditions at municipal waste dumps [22]. The process of transformation of bio-organic substances by the methanogenic community is accompanied not only by the accumulation of biogas but also by the gradual formation and accumulation of HSs in the non-gas phase [20]. The HSs that accumulate in this case lead to the inhibition of methanogenesis [98], which is why it is important to control the appearance and concentration of HSs in the anaerobic digestion of various pollutants, including SPs.

Under anaerobic conditions, SPs are usually not the target substrate of the process but get into the reactor as part of the transformed waste or as a carrier for MOs, which act as anaerobic digestion biocatalysts (Table 6 [98,99,100,101]).

During the anaerobic transformation of processed activated sludge dewatered with cationic RAM, it was found that, unlike the processes described earlier, in this case, the SPs residues were bound not to HSs but to amino acids on the surface of activated sludge cells. The authors of this study explained the preferential binding of RAM to amino acids, rather than to HSs, by the superior initial protein content in the sludge compared to the initial HS content at the beginning of the process [99].

Poly (vinyl alcohol) cryogel was used as a carrier matrix for a methanogenic consortium [100,101]. For more than three years, such an immobilized biocatalyst was used in the processes of anaerobic transformation of sulfur-containing waste from oil desulfurization, during which HSs were accumulated in the liquid phase as anaerobic digestion products. During this period of time, the biocatalyst, which was in the reaction medium, completely retained its integrity. A similar biocatalyst was used in the process of biogas accumulation in the presence of natural and chemically modified HSs at a concentration of 1–10 g/L [98]. The integrity of this biocatalyst was also preserved without degradation of the polymer carrier.

In general, anaerobic processes with a low level of oxidative potential should not be considered processes that are attractive and effective for the degradation of SPs. Additional confirmation of this is provided by the previously presented data on the increasing concentration of MPs deep into bottom sediments, which are not subject to or are subject to a lesser degree of transformation due to the absence of aerobic conditions.

Thus, it can be summarized that HSs are present simultaneously with SPs in various environmental objects and perform different functions in the process of their biodegradation. According to the theoretical analysis of the conducted research works on the search for microbial biodestructors of SPs, the pool of MOs that are susceptible to the influence of HSs (Table 1, Figure 2) significantly exceeds the pool of those MOs that were used in the conducted studies on the biodegradation of SPs in environmental objects (Table 2, Table 3, Table 4 and Table 5) in the presence of HSs. This difference can be considered a huge potential revealed in this review and used as an incentive for further research works with the control of concentrations and characteristics of HSs involved in such processes.

## 3. Analysis of Mutual Participation of HSs and MOs in the Processes of Biodegradation of SPs

Analysis of literature data (Table 1, Table 2, Table 3, Table 4, Table 5 and Table 6) allows concluding that, in general, HSs have a multifaceted effect on the degradation of SPs by MO cells (Figure 1). The mechanisms of mutual influence of HSs, SPs, and MOs have still been studied very superficially (not deeply). It is known that HSs can enter into various donor–acceptor interactions with living cells and substances of organic nature. HSs as electron shuttles can be the basis for the development of new approaches to the biodegradation of SPs [14]. HSs stimulate the synthesis of MO enzymes, which are involved in the processes of cell adhesion to SPs and their destruction.

Analysis of the data in Table 1, Table 2, Table 3, Table 4, Table 5 and Table 6 allowed us to identify several main ways in which HSs participate in the biodegradation of SPs. In particular, HSs can have both a direct effect on the properties of polymers (sorption of HSs on the surface of SPs, modification and change in the surface properties of the polymer, which contribute to an increase in the bioavailability of SPs for cells) and an indirect participation through enhancing the metabolic activity of MOs, stimulating an increase in the number of their extracellular enzymes involved in biodegradation and maintaining oxidation processes. HSs can bind to SPs of various structures mainly through hydrophobic, electrostatic, Van der Waals interactions and the formation of hydrogen bonds [73,74].

In preparing this review, the authors estimated the degradation velocity for the most common polymers in the presence of HSs from various sources (Figure 3) based on the data presented in Table 2, Table 3, Table 4, Table 5 and Table 6.

The data presented in the original articles were used to calculate the degradation rates. It is not always possible to reliably determine which contribution to a given value is made by abiotic factors and which is biotic. Therefore, their overall contribution was taken into account. Since different authors controlled the process of polymer destruction by different methods, calculations were carried out as a percentage of the controls in the conducted studies. In general, the range of calculated polymer degradation rates is in the range of 0–0.5% per day, with the exception of data obtained during the degradation of polyacrylamide and polycaprolactone. Moreover, in the case of polyacrylamide, this is due to the characteristics of the polymer itself (it is easily biodegradable). In the case of polycaprolactone, it was the conditions of separate composting that allowed for high results. As a result of the calculations, the degradation velocity of conditionally biodegradable polycaprolactone can be arranged in the following order of decreasing this parameter under environmental conditions: compost > natural water sources > soil (Figure 4). These results correlate with the HS content in these media (up to 250 > up to 200 > up to 120 g/kg) [20].

High rates of biodegradation during polymer composting are due not only to the presence of HSs in the compost but also to the microbial composition of those consortia that are characteristic of compost. Here, a significant number of mycelial fungi are noted, possessing a significant pool of oxidative and hydrolytic enzymes responsible for the degradation of polycaprolactone [102]. The bottom sediments of natural reservoirs also contain a significant amount of humic substances (as noted above), and there is a significant pool of MOs in the consortium. However, these MOs are anaerobes, which are characterized by lower metabolic rates.

For conditionally non-biodegradable polyethylene, the values of degradation velocity in the different media are arranged as follows: water > soil > compost. Probably, in the case of polyethylene degradation, the abiotic component makes a significant contribution to this process, which was noted earlier. In the near-surface layer of reservoirs and soils, the contribution of abiotic factors to the decomposition of synthetic polymers is maximal. UV light does not penetrate into the compost layer, and the mechanical component caused by the movement of the layers is minimal. On the other hand, during composting, MOs play a major role in degradation, as well as the thermal phase of composting.

The results of the calculation allowed us to draw several conclusions. For example, polyacrylamide decomposes faster in soil than as a result of anaerobic digestion. This conclusion seems logical since processes in anaerobic conditions always proceed more slowly than in aerobic ones. Nevertheless, anaerobic digestion is possible, for example, during the decomposition of wastewater sludge, including polymer particles. Composting, in general (Figure 4), seems to be an attractive method for the destruction of various SPs. It is under these conditions that, for example, the enzyme cutinase [103,104] and other hydrolytic enzymes, which are now being studied by many researchers as biocatalysts for the destruction of a number of SPs, especially polyethylene terephthalate hydrolase of filamentous fungi, exhibit high catalytic activity [102].

It is obvious that the most favorable conditions are created in compost for the manifestation of enzymatic activity by various MOs, including those found in polymicrobial consortia with fungi [105]. In addition, the concentration of HSs in the process of composting constantly increases as it matures in contrast, for example, to the soil. The soil as a medium for the destruction of SPs is in second or third place in the indicated series of biodegradation rates, where consortia of fungi and bacteria are also present. The element composition of HSs in compost differs from the composition of those HSs that are present in the soil (Table 7 [17,87,90,106,107,108,109,110,111,112,113,114,115,116,117,118]). Apparently, the processes of mineralization and humification occurring during composting can be controlled and enhanced by introducing various components and MOs [20]. The development of research in this direction may become the key to solving problems related to the possibilities of rapid and complete decomposition of SPs. Following this idea, in this review, the ratios of chemical elements in HSs participating in the processes of biodegradation of SPs were analyzed (Table 7).

The H/C ratio is an important indicator determining the aromaticity of HSs. Higher atomic H/C ratios correspond to lower aromaticity [20]. The high aromaticity of HSs ensures their stability under environmental conditions. This characteristic is possessed by HSs isolated from soil; they can retain their properties during a fairly long process of biodegradation of SPs and have a stable effect on the metabolism of MOs for a long time.

The high C/N ratio found in HSs corresponds to a significant content of organic matter. This ratio is key to ensuring favorable conditions for the functioning of MOs. The optimal value of this parameter is 20–40. A C/N ratio of less than 10 leads to the suppression of MO activity [22]. For the biodegradation of SPs, in terms of the C/N ratio, it is most appropriate to use HSs from compost or the upper soil horizon.

The O/C ratio in HSs reflects the proportion of oxygen-containing functional (carboxyl, hydroxyl, and carbonyl) groups in combination with aromatic structures. It serves as evidence of the ability of HSs to enter into donor–acceptor interactions, form hydrogen bonds, and actively participate in the processes of sorption of SPs. Such characteristics of HSs are important for the sorption and change in the surface properties of polymer particles and, thereby, for the initial stage of their biodegradation. HSs obtained from bottom sediments or waters of natural reservoirs are characterized by the maximum value of this indicator.

Thus, high values of the C/N (around 20–40) and O/C (more than 0.6) ratios in the composition of HSs, in general, should contribute to an increase in the efficiency of biodegradation of SPs when the first indicator is important for the functioning of MOs, and the second indicator is responsible for the sorption of HSs on the polymer, donor–acceptor interactions between SPs and HSs, and the formation of hydrogen bonds. These findings explain the best indicators for the rates of destruction of SPs in compost and water bodies (Figure 4). This information can be used to develop eco-friendly processes for the biodegradation of SPs.

## 4. Effect of HS Photochemical Activity on PS (Bio)Degradation

It is known that HSs actively absorb solar radiation and exhibit photochemical activity [119], which can affect the toxicity, mobility, and bioavailability of substances located near them, changing their molecular structure, participating in the mineralization of organic carbon to CO due to photodegradation. It is known that HSs adsorbed on SPs or located with them in the same environment and available for UV light (surface layers of soils and compostable materials, upper layers of natural water sources) can intensify their biodegradation or, conversely, inhibit this process due to their participation in photo-oxidative processes (Figure 5, Table 8 [15,120,121,122,123,124]).

In vitro experiments in the presence of HA at pH 7.0 showed accelerated surface oxidation of polypropylene–MPs [120] and enhanced photoaging of poly (vinyl chloride)-MPs [121]. Acceleration of photodegradation of polystyrene microplastics (polystyrene–MPs) in an aqueous environment with HA was found due to the formation of large amounts of ROS (0.631 mM •OH), which contributed to the destruction and degradation of the polymer chain and modification of its surface structure [15]. Other researchers noted that photoaging of polypropylene–MPs was significantly slowed down in lake water compared to ultrapure water after 12 days of UV irradiation. In this case, HA and FA were identified as the main factors contributing to the observed inhibition.

HSs acted as both ROS scavengers (e.g., of •OH) (dominant contribution) and optical light filters. As ROS scavengers, HA and FA significantly decreased the capacity for the formation of •OH and O_2_•^−^ by polypropylene–MPs under irradiation. In addition, the chromophores in HA and FA competed for photons with MPs through the light-shielding effect, thereby causing less fragmentation of polypropylene microparticles [123].

In a comparative analysis of the effects of HA and FA on the degradation process of polyethylene–MPs for 15 days using artificial irradiation, it was found that HA promoted the photo-oxidative degradation of polyethylene–MPs with a higher weight loss value and the production of particles with a smaller average size (110 μm) compared to the same effect of FA. It was found that HA has a higher ability to generate free radicals (•OH and O_2_^−^) than FA, which accelerates the photo-oxidation of polyethylene–MPs and leads to different mechanisms for the implementation of photodegradation of polymer particles [125].

Despite the fact that the presence of HSs promotes the formation of ROS, it should be noted that HSs themselves have a certain antioxidant potential [98]. It was shown that the introduction of ammonium salts of HA into the composition of poly(vinyl alcohol) at a concentration of 0.5–5% contributed to an increase in the stability of the polymer. However, sodium salts of HA at a concentration of 10% had a destabilizing effect.

In all samples of media with SPs, where polymer degradation had already begun, the presence of HA and their salts caused more intense destruction [126]. Today, HA is actively proposed as individual [15,120] or as part of hybrid composite [122] catalysts accelerating the degradation of polymers, in particular, MPs, which should be added to the treated media from the outside. Thus, the synthesis of hybrid composites, in which an inorganic oxide with semiconducting properties (i.e., TiO_2_, ZrO_2_, or ZnO) can be combined with an organic molecule to form ligand–metal charge transfer complexes, is a promising strategy leading to the production of hybrid nanomaterials capable of stabilizing and/or producing ROS [14,127]. In particular, ZnO nanoparticles with HA were successfully tested as photocatalysts for the degradation of linear low-density polyethylene and polylactide thin films under UVA/light irradiation [122].

Interestingly, the efficiency of SP degradation in the near-surface soil layers depends little on the total dose of solar radiation and temperature, but the soil composition, primarily the concentration of dissolved organic matter and water concentration, significantly affected the rate and extent of polymer photo-oxidation [124]. HA and FA, which are part of the soil-dissolved organic matter, contributed to an increase in the rate of polyethylene photo-oxidation due to the formation of ROS [124].

Apparently, in the case of degradation of SPs in soil and open composting systems, it is advisable to take into account hybrid processes combining photo-oxidation and biocatalytic transformation of plastics under the action of MOs in the presence of different HSs.

When studying the processes of SP destruction in the presence of HSs in real conditions, in addition to photocatalytic properties, it is certainly necessary to evaluate the effect of the possible manifestation of other properties of HSs. Thus, it was noted that the filtrate after the decomposition of poly(vinyl chloride)-MPs in the presence of HA exhibited higher acute and genetic toxicity when irradiated with UV light compared to dark conditions of polymer processing.

Thus, when assessing the environmental risk from the presence of SPs in surface waters, it is necessary to take into account the influence of the present and formed potentially toxic soluble organic components from polymers and the composition of HSs, which are common photosensitizers in nature and promote the transformation of various organic compounds under the action of ROS formed under the influence of sunlight [121]. Moreover, it is obvious that the ability of HSs to generate ROS gives them antimicrobial properties [76], which increase with an increase in the concentration of HSs [22]. However, the found balance between the concentrations of HSs producing ROS under natural conditions and the degradation activity of MOs in the destruction of SPs can give significantly improved results. Thus, complex pathways for the formation of •OH radicals from HA were studied, which influenced the process of transformation of polystyrene–MPs. It was found that •OH radical could be generated continuously during successive redox cycles of HA by *Bacillus thermotolerans* cells, and •OH generated as a result of redox cycles with HA could lead to 18.1% weight loss of polystyrene–MPs within 8 weeks [90].

It is interesting that HA is not the only source of ROS formation under the influence of sunlight, but also some exopolysaccharides secreted by microalgae, for example, *Chlorella vulgaris* cells, widely used in biological water purification [75]. Thus, an increase in the aging of polystyrene–MPs (3 μm) was shown in a medium with exopolysaccharides secreted by *C. vulgaris*, under simulated sunlight for 20 days. The presence of exopolysaccharides led to the appearance of polystyrene particles smaller than 1 μm in size, as well as the formation of hydroxyl groups on the surface of polystyrene particles. In addition, exopolysaccharides contributed to the formation of dissolved organic matter from polystyrene, which was similar to the photosensitizing (carbonyl and hydroxyl) groups of HA [75]. It should be noted that *C. vulgaris* cells survived under these conditions in the presence of HSs and displayed their characteristic properties, which contribute to the intensification of MP decomposition processes.

Thus, of the abiotic factors influencing the degradation process of SPs, the main effect is probably UV radiation. Under environmental conditions, photo-oxidation and biodegradation of SPs in the presence of HSs are closely related processes (Figure 4). ROS formed during photo-oxidation, and the triggered polymer oxidation processes can have a negative effect on living MOs cells. HSs can both initiate the process of ROS formation under the influence of sunlight on the surface of water or soil and trap ROS, preventing their toxic effect on MO cells, including in the thickness of the soil/reservoir, etc., where sunlight is insufficient. The composition of HSs and their concentration are extremely important factors in such biosystems.

## 5. Prospects for the Development of Processes Associated with the Presence of MPs, MOs, and HSs in Ecosystems

In environmental conditions, all the above components of ecosystems are found to be tightly intertwined (Figure 6). Plants can also be added to them, the cultivation of which is carried out on soils, including those containing MPs, the source of which in these environments can be atmospheric precipitation [128].

Plants and MOs are capable of entering into a close symbiosis and regulating each other’s metabolism. HSs, in turn, also affect plant development and the vital activity of MOs. MPs absorb HSs, change soil characteristics, and inhibit plant growth. Similar interactions exist in the aquatic environment. MPs negatively affect the photosynthetic apparatus of microalgae [86], which are capable of effectively destroying SPs. For example, polyethylene film, which is traditionally used for mulching during plant cultivation, reduced the ability of the soil to accumulate C, N, P, and HSs [129]. The presence of polyethylene mulching film in the soil led to an increase in the biomass of native bacteria but reduced the diversity of the bacterial community. As a result, a decrease in the biomass and yield of rice by 11.34 and 19.24%, respectively, was noted. Thus, the presence of the polymer had the most significant negative impact on the nutrient supply in the soil, followed by changes in the microbial community structure and metabolic functions of MOs, which affected crop yield.

Some rhizosphere MOs are able to form biofilms on SPs and intensify their biodegradation. Such MOs include the bacteria *Serratia plymuthica* and the fungi *Laccaria laccata*. [108]. For example, a combination of *Miscanthus giganteus* and the rhizosphere fungus *L. laccata* was used for the degradation of polylactide and polyethylene terephthalate. This combination accelerated the biodegradation of SPs in compost soil. A significant increase in biomass was observed for plants in such an environment.

In an aquatic environment, the “main impact” of the presence of SPs is borne by MOs, primarily phototrophs, which experience a deterioration in the functioning of their photosynthetic apparatus. Larger MP particles impede the penetration of light in an aquatic system and, thus, inhibit photosynthesis. Smaller MP particles are adsorbed on the surface of microalgae cells and destroy their cell wall [130]. HSs in such systems bind smaller MP particles and reduce their negative impact on cells, whereas, in the case of larger MP particles, this effect is not observed [130]. In order to reduce the toxic effect of MPs on microalgae, it is necessary to introduce HSs with predominantly negatively charged functional groups into the system. This allows them to bind to MP particles and, thus, prevent the latter from being adsorbed on exopolysaccharides and the surface of microalgae [86].

Cyanobacteria are able to adapt to the negative impact of MPs quite quickly (~14 days) [131]. It was noted that the content of humic-like substances in the exopolysaccharides of cyanobacteria increases during such adaptation, which probably reduces the toxicity of polymers for phototrophs [131]. Interestingly, increased secretion of humic-like compounds in the exopolysaccharides of microalgae was noted precisely in the presence of MPs in the environment [132]. The presence of SPs in high concentrations in the aquatic environment contributes to an increase (after preliminary adaptation) in the biomass of cyanobacteria [131]. In turn, the accumulated biomass of phototrophs flocculates and sediments MPs of various polymers into bottom sediments, which leads to a change in the chemical composition of dissolved organic matter in water bodies [133].

The resulting small molecules of HSs and humic-like substances, in turn, can penetrate into cells and trigger their “self-defense” processes, which leads to the appearance of ROS as a response and increased synthesis of exopolysaccharides by cells [86].

In certain cases, materials made from some SPs are purposefully introduced into various environmental objects. The most common case of purposeful introduction of polymers into soil is associated with giving the soil a certain structure, filling voids, and protecting it from erosion. For this purpose, the use of polyacrylamide [134], poly (vinyl alcohol) cryogel [135,136], and polyurethane [137] is proposed. Such additives also help to retain moisture and maintain constant soil moisture regardless of weather conditions, which is especially important for plants. Thus, poly (vinyl alcohol) cryogel was used as a carrier for immobilizing cells capable of biodegrading oil pollutants in soil [138,139]. The resulting biocatalyst was introduced (buried) directly into the soil and used for 14 months. Poly (vinyl alcohol) cryogel is also introduced into water bodies for the sorption of heavy metal ions without the destruction of SPs [140].

Some studies have noted that MPs that enter the soil play a dual role. On the one hand, they stabilize the structure of some types of soil and promote more active plant growth. On the other hand, they can suppress and change the composition of native natural consortia [141,142]. Thus, SPs (in particular, MPs), when entering ecosystems, ultimately lead to an increase in the total biomass of adapted native cells of different MOs. This, in turn, leads to difficult prediction of events in the long term.

Interactions between MOs, HSs, and environmental factors in the degradation processes of SPs are diverse, interesting, and accompanied by different effects that directly or indirectly affect the destruction of SPs. It is important to highlight the following key points, which reflect the positive and negative aspects of the interactions discussed above:-HSs exhibit surfactant properties that modify the surfaces of MP particles, thereby improving their accessibility to enzymes responsible for the biodegradation of polymers;-Processes of adhesion/immobilization of MOs on HSs are known, while the possibilities for the formation of more numerous populations by cells with the manifestation of quorum effects are realized. As a result, the efficiency of biocatalytic processes and the resistance of cells to the negative impact of external factors increase [72];-There is a direct relationship between the presence of low concentrations of HSs and increased microbial respiration, an increase in the concentration of biofilm biomass per unit area of polymers, and stimulation of the synthesis of microbial enzymes that are involved in the processes of MP destruction;-HSs provide MO cells with protection from MP particles and xenobiotics due to competitive adsorption interactions, prevent the formation of toxic MP complexes with xenobiotics, and improve the availability of pollutants for biodegradation, promoting the intensification of the SPs biodegradation process;-HSs act as additional sources of nutrients for MOs, which contributes not only to the growth of MO cells in the ecosystem but also to an increase in their diversity in situ. HSs, even in the presence of MPs, stimulate the restoration and dynamic development of microbiological communities, but this process requires special control;-In an aqueous environment, HA promotes the formation of ROS, which has a positive effect on the processes of photoaging and surface oxidation of SPs, while FA can act as ROS “traps”;-HSs bind to MP particles through the formation of hydrogen bonds and hydrophobic interactions, and in water bodies, such complex particles can sediment to the bottom, and in soil, they can change the characteristics of the environment, which, as a consequence, can lead to inhibition of plant growth;-HSs gradually accumulate in anaerobic bioreactors used for methanogenesis and processing of sludge waste from treatment plants and inhibit methanogenesis, while HSs do not always exhibit good sorption properties in relation to MPs;-HSs can enter into various donor–acceptor interactions with living cells and organic substances, on the basis of which HSs as electron shuttles can be the basis for the development of new approaches to the biodegradation of SPs, the nature, composition, and concentration of HSs affect the efficiency of polymer biodegradation under the influence of different MOs.

## 6. Conclusions

In fact, this is the first review that attempted to find the relationship between different elements of ecosystems (HSs, SPs, MOs) and the processes of interaction between them in order to assess the possibility of predicting the development of such ecosystems and managing them. At the moment, it is clearly difficult to accomplish this since the system is multifactorial. However, the calculated values of degradation velocity of SPs in the presence of MOs and HSs, made in this work, revealed the dependence of this parameter on the types of SPs, the presence of water, and additionally introduced HSs. The processes are long enough, but to accelerate them, subsequent targeted accumulation of information is needed, which can attract new researchers to this topic and form the development of a separate scientific direction. But it can be certainly stated that the pools of information on MOs as SPs destructors and on the possibility of regulating microbial metabolism by changing the concentrations and characteristics of HSs present or introduced into ecosystems open up significant prospects for further research.

## Figures and Tables

**Figure 1 microorganisms-12-02024-f001:**
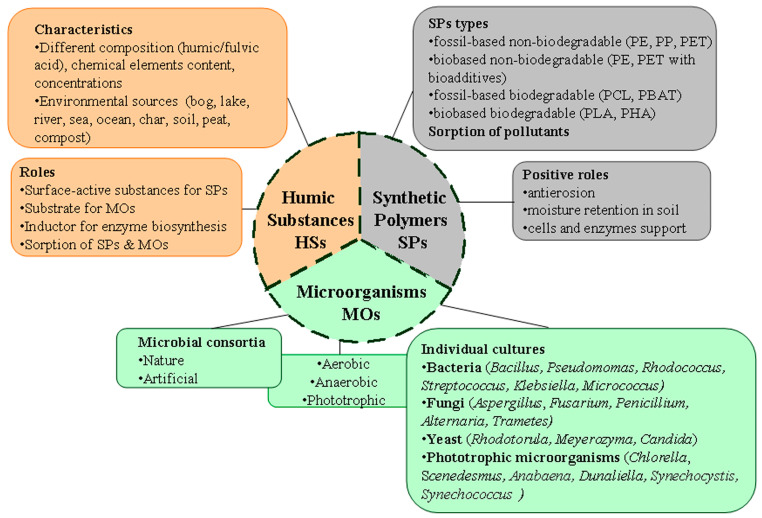
The main objects of analysis in this review and their characteristics are important for mutual influence and the provision of multidirectional effects on the (bio degradation of SPs.

**Figure 2 microorganisms-12-02024-f002:**
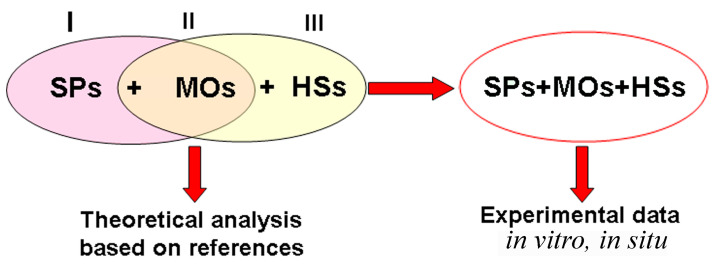
Principal scheme of theoretical analysis of biodegradation of synthetic polymer in the presence of HSs undertaken in this review: I, II—information about biodegradation efficiency of SPs by various MOs, and III—information about influence of HSs on MOs with corresponding references [25,26,27,28,29,30,31,32,33,34,35,36,37,38,39,40,41,42,43,44,45,46,47,48,49,50,51,52,53,54,55,56,57,58,59,60,61,62,63,64,65].

**Figure 3 microorganisms-12-02024-f003:**
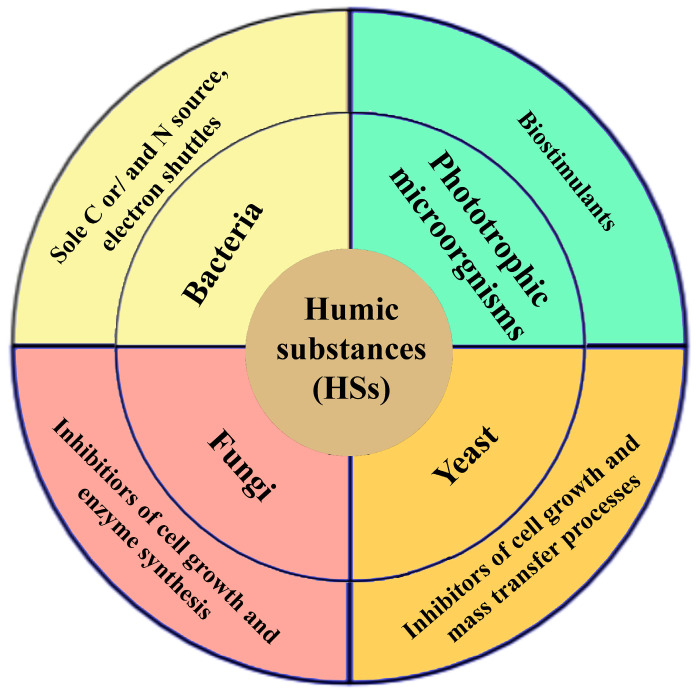
The main role of HSs for different microbial cells (picture is based on the information from references [25,26,27,28,29,30,31,32,33,34,35,36,37,38,39,40,41,42,43,44,45,46,47,48,49,50,51,52,53,54,55,56,57,58,59,60,61,62,63,64,65]).

**Figure 4 microorganisms-12-02024-f004:**
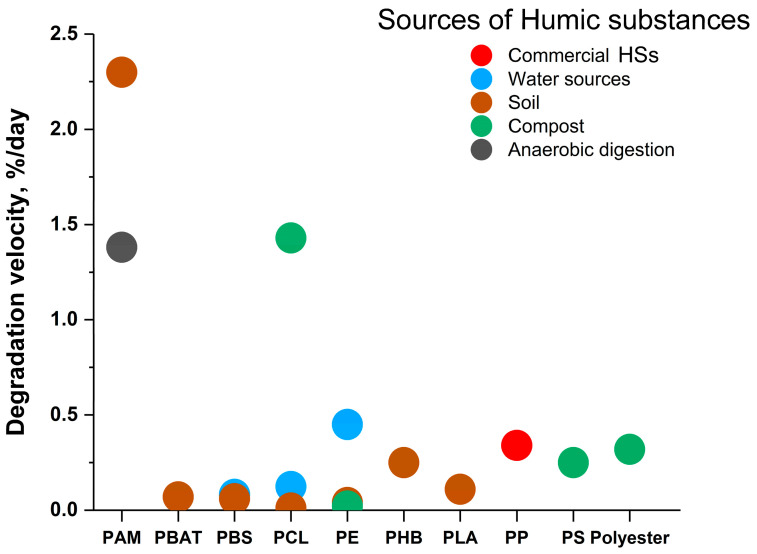
Calculated values of degradation velocity of SPs in presence of MOs and HSs (based on the data of Table 2, Table 3, Table 4, Table 5 and Table 6).

**Figure 5 microorganisms-12-02024-f005:**
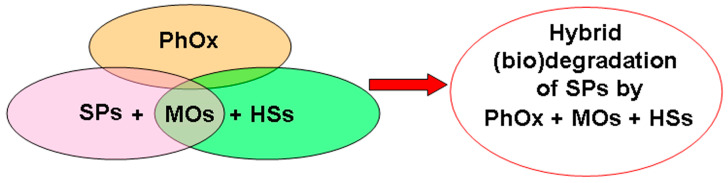
The influence of photo-oxidation (PhOx) on the processes of (bio)degradation of SPs in presence of HSs and MOs.

**Figure 6 microorganisms-12-02024-f006:**
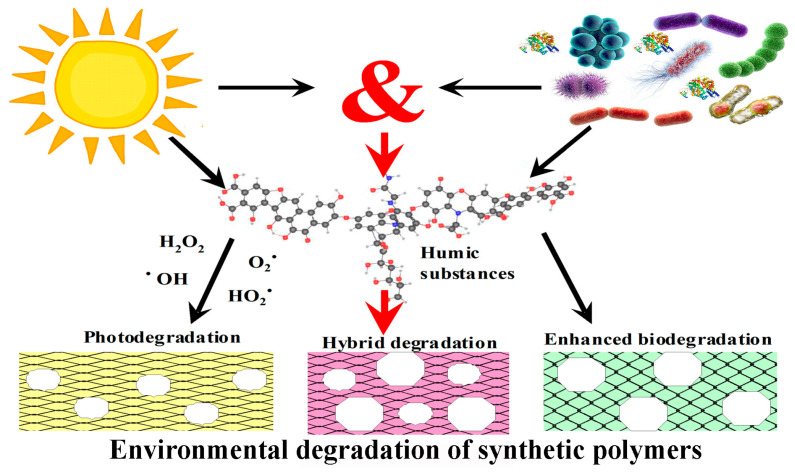
The interactions between HSs, MPs, and MOs under environmental conditions with the participation of PhOx.

**Table 1 microorganisms-12-02024-t001:** MOs capable of biodegrading SPs whose metabolism is influenced by HSs (color of columns corresponds to Figure 2).

* I. SPs/Size/Time/% of Degradation (Weight Loss) [Reference]	II. MOs	III. HSs and Their Effect on the Mos [Reference]
**Bacteria**		
PET (212–500 µm)/18 days/49.2–68.8% [25]	*Streptomyces* sp.	HSs are used as the sole carbon source [54]
PE (MW 5000–10,000 DA)/15 days/13.0–17.3% [26]	*S. albogriseolus*	
PS (1.5 × 1.5 cm)/60 days/34% [27]	*Bacillus paralicheniformis*	HSs increase the bacterial growth rate and are used as source of N [55]
LDPE (1 × 1 cm)/180 days/18.9% [28]	*Bacillus* sp.	
PVC (3 cm^2^)/30 days/19% [29]	*Pseudomonas citronellolis*	HSs increase the bacterial growth rate [56]
PE (1 × 1.5, 1.5 × 2 cm)/56 days/6% [30]	*P. knackmussii*	
PS/15 days/2.6% [31]	*Pseudomonas* sp.	
LDPE (3 × 2 cm)/30 days/11.2% [32]	*Rhodococcus* sp.	The stimulating effect of HSs on bacterial growth, HSs are used as a carbon source [57]
PP (2.4 mm)/40 days/6.4% [33]	*Rhodococcus* sp.	
EVA (0.2 × 0.2 or 2.5 × 2.5 cm^2^)/30 days/0.65% [34]	*Klebsiella aerogenes*	HSs can act as electron shuttles to promote electron transfer from extracellular respiratory bacteria to solid electron acceptors or organic pollutants [58]
PE (1 × 1 cm)/30 days/2.2% [35]	*K. pneumoniae*	
HDPE (2 × 2 cm)/90 days/3.9% [36]	*Micrococcus luteus*	The MOs can use HSs as the sole carbon source and energy [59]
PVC (3 cm^2^)/180 days/8.4% [37]	*Micrococcus* sp.	
**Fungi**		
PU (2 × 2 cm)/25 days/15–20% [38]	*Aspergillus fumigatus*	Inhibition of cell growth and enzyme synthesis [60]
PVC (10 × 2 cm)/30 days/2.2% [39]	*A. fumigatus*	
PP, PE, PS (3–5 mm)/70 days/18.3%, 6.8%, 1.9% [40]	*A. flavus*	
PP, PE, PS/70 days/6.7%, 5.1%, 3.3% [40]	*A. versicolor*	
PVA/10 days/81% [41]	*Penicillium brevicompactum*	Inhibition of cell growth [61]
PHB and PHBV/7 days/99% [42]	*P. oxalicum*	
LDPE (2 × 2, 5 × 5 cm)/90 days/1.5–1.7% [43]	*Fusarium oxysporum*	Inhibition of growth up to 30% at HSs 50 mg/kg soil [62]
PU (thickness of 0.1 mm)/60 days/3.3% [44]	*Alternaria alternata*	No growth inhibition in presence of 200 mg/L of HSs [63]
**Yeast**		
PET (75–300 μm)/30 days/10% [45]	*Vanrija* sp.	Growth inhibition: HSs compose an additional layer on the yeast surface and decrease diffusion of nutrients and metabolites [64]
UV-irradiated 13C-labeled PE/365 days/3.8% [46]	*Rhodotorula mucilaginosa*	
PE (10 × 3 cm)/60 days/13.9% [47]	*Meyerozyma guilliermondii*	
LDPE (5 × 3 cm)/49 days/3.2% [48]	*Candida tropicalis*	
**Photosynthetic micro-organisms**		
LDPE (1 × 1 cm)/30 days/8.2% [49]	*Anabaena spiroides*	Biostimulants [65]
LDPE (1 × 1 cm)/30 days/3.7% [49]	*Scenedesmus dimorphus*	
PET (2 × 2 cm)/30 days/5.5% [50]	*Chlorella vulgaris*	
LDPE (2 × 2 cm)/45 days/20.2% [51]	*Picochlorum maculatum*	
LDPE (1 × 1 cm)/42 days/30% [52]	*Oscillatoria subbrevis*	
LDPE (1 × 1 cm)/42 days/27% [53]	*Nostoc carneum*	

* **PU**—polyurethane, **EVA**—ethylene vinyl acetate, **PHB**—poly(3-hydroxybutyrate), **PHBV**—poly(3-hydroxybutyrate-co-3-hydroxyvalerate), **PP**—polypropylene, **PS**—polystyrene, **PVA**—poly(vinyl alcohol), **PE**—polyethylene, **PVC**—poly(vinyl chloride), **PET**—polyethylene terephthalate, **HDPE**—high-density polyethylene, **LDPE**—low-density polyethylene.

**Table 2 microorganisms-12-02024-t002:** The effect of HSs on biodegradation of SPs in vitro (lab scale).

SPs (Sample Size) [Reference]	MOs, Type of HSs, Conditions of Biodegradation	Result
PP (≥1 mm) [73]	4 g/L epilithon biofilms (*Proteobacteria*, *Actinobacteria*, *Bacteroidetes*) and HA (5 mg/L, Sigma-Aldrich, St. Louis, MO, USA), 30 °C, in the dark, 36 days	Weight loss of PP —12.3%
PS (80 nm) [74]	*Chlorella vulgaris*; amino-functionalized polystyrene nanoplastics (0.05–0.4 mg/L) and atrazine (10 μg/L) with HA (1 mg/L, Sigma); cultivation at 23 °C, with white fluorescent lamps, 21 days	HA protected the *C. vulgaris* cells from oxidative stress induced by atrazine present on the surface of PS
PS (3 µm) [75]	*Chlorella vulgaris*, 2000 lx (light: dark = 18:6), 25 °C, Xenon test chamber (1.8 kW xenon arc lamp) and a Daylight-Q filter-0.1 g of PS/20 mL of the extracellular polymeric substances (EPS) extracted from microalgae biomass (pH 6.8; 7.0 mg/L dissolved organic carbon); irradiance-0.68 W/m^2^ (340 nm), 20 days	The PS surface became rough and exhibited a layered peeling. The increased OH production can be attributed to the presence of HA-like substances in the cell EPS
Hydroxypropyl methyl cellulose (film) [76]	A from lignite was introduced to the polymicrobial biofilm. Biodegradation was in presence of fungi (*A. niger*, *P. funiculosum*, *P. variotii*, *A. terreus*, *A. pullulans*, *P. ochrochloron* —10^6^ spores/mL for each type of cells), 10% (*w*/*w*) HA, 6 months	Hydroxypropyl methyl cellulose film biodegradation was 91%.

**Table 3 microorganisms-12-02024-t003:** The effect of HSs on biodegradation of SPs in water ecosystems.

* SPs (Sample Size) [Reference]	MOs, Type of HSs, Conditions of Biodegradation	Result
PE (1.2–40.0 μm) [84]	HSs-containing lake water (Lake Haukijärvi, dissolved organic carbon-16 mg/L, pH 6.4), 2 g PE in 150 mL lake waters, 56 days, 110 rpm, 21 °C	Degradation rate of PE-0.45 ± 0.21% per year. The biodegradation rate of PE in the HS-containing lake water was 5–30 times higher than in the clean lake water.
PLA, PET, PP-100~150 μm [85]	The NOR adsorption experiments: 0.1 g aged MPs (400 mM K_2_S_2_O_8_, 70 °C, 12 h) and 20 mL of NOR solutions with different concentrations (1, 2, 4, 6, 10, 15 mg/L), 25 °C, 36 h with addition of HA (0–8 mg/L, Analytical pure Shanghai, China)	HA had an inhibitory effect on the adsorption of NOR on the MPs which may be caused by the competitive adsorption interaction between HA and MPs.
PCL (0.2–3 mm), PBS (0.1 mm), PBAT (0.5 mm) [24]	Seawater, 20 °C, 6–12 months; plastic particles were enclosed in a larger fishing net and submerged in coastal seas (Republic of Korea) at a depth of 1.5–2 m, 12 months	PCL weight loss in aquarium with sea water—0.2 mm samples fully degrade after 6 months, 2 mm samples—20% after 1 year; in real marine seawater PCL (2–3 mm)—35–45% degradation; PBS and PBAT > 30%

* **PLA**—polylactide, **PCL**—polycaprolactone, **PBS**—polybutylene succinate, **PBAT**—polybutylene adipate terephthalate.

**Table 4 microorganisms-12-02024-t004:** The effect of HSs on biodegradation of SPs in soil.

* SPs (Sample Size) [Reference]	MOs, Type of HSs, Conditions of Biodegradation	Result
LDPE–1 × 2 cm [87]	Communities of MOs isolated from the LDPE surface incubated in the Orthic Acrisol soil (*Proteobacteria*, *Bacteroidetes*, *Actinobacteria*) with 0.05–0.50 mg/mL HA (extracted from the upper horizons of ferrallitic soil, typical chernozem and humate fertilizer based on lowland peat), 30 °C, 14 days	Increasing roughness of the LDPE surface by 2.0–3.5 times
PLA, PHB, PBAT (5 × 5 mm), PE (0.5–2 mm) [88]	Lufa 2.2 soil—1.8% organic carbon, 0.02% nitrogen, pH 5.6; 300 mg of plastic, 30 g soil (dw), and 15 mL of mineral media, 12 months. Carbon content—49% and 42% for FA and HA, respectively	Mineralization of PBAT; degradation: PLA—25%, PHB > 90%, PE—15%
PAM–(3 × 10^6^ g/mol) [17]	Soil (11.5 g/kg organic matter, pH 6.9) with 0.07 g/g immobilized on biochar (sewage sludge and coconut shell) *Klebsiella* sp. (0.5 × 10^9^ CFU/g), 1.5 mg PAM/g dry weight soil, pH 6.6, 38 °C, 30 days	Degradation—69.1%.
PCL, PBS, PBAT (films 0.2 mm–2 mm thickness and size 3 × 3 cm or powder 200 μm) [24]	Horticultural topsoil and fertile soil (140 g of topsoil, or 98 g of topsoil, and 42 g of vermicompost (fertilized soil) mixed with 140 g of water), 6 months	Weight loss: in the horticultural soil: PCL—2%, PBS—1%, PBAT—0.2%; in the fertilized soil–PBAT—0.2%, PCL and PBS—100%
DBP [89]	20 mg/kg of DBP in mollisol soil (Harbin, China, pH 7.8, organic matter 2.75%) with addition of 1.5 g/kg HA (Shanghai Ryon Biological Technology Co., Ltd., Shanghai, China), 60 days	The half-life of DBP was shortened from 11.65 to 3.36 days

* **PAM**—polyacrylamide, **DBP**—dibutyl phthalate.

**Table 5 microorganisms-12-02024-t005:** The effect of processes with participation of HSs on biodegradation of SPs in composting.

* SPs (Sample Size) [Reference]	Processes with Participation of HSs, Conditions of Biodegradation	Result
PS-MPs (5 × 5 mm) [90]	*Bacillus thermotolerans*, aerobic process, 6 h, tryptic soy broth and anaerobic process, 18 h, minimal salts medium supplemented with 5 mM Glucose as an electron donor, 50 °C. Commercial humic acid (CHA—Tianjin Guangfu Fine Chemical Research Institute), chicken manure compost humic acid (CMHA), dairy manure compost humic acid (DMHA), and sludge compost humic acid (SHA)—10 g/L of medium	Oxidative degradation during HA redox cycles—weight loss of PS-MPs—5.9–18.1% within 8 weeks The lowest efficiency of PS-MPs degradation was in SHA than in DMHA, CMHA, and maximum with CHA
PP and PP composites (50 × 120 mm) [91]	Abiotic treatment (8 h of UV exposure at 60 °C and 4 h of condensation at 50 °C) and composting 58 °C, 45 days	Degradation—11.2–36.4%
PCL/starch (60/40 *w*/*w*)—50 μm Polyester—60 μm PE with peroxidant—36 μm [92]	Composting, 50 days	Biodegradation for material PCL/starch—88%, polyester—60%, PE—1.1%
Acrylic acid-grafted PP—80–85 μm [93]	Composting, 58 °C, 45 days	Carbon conversion from SPs to CO_2_—1.5–5.6%
LDPE–starch (0–50%)—4 × 4 cm [94]	Composting, 125 days	Weight loss—0.25–13.03%
Copolyester poly (succinate-co-glutarate-co-adipate-co-terephthalate 1,4-butylene)—10 × 10 cm [95]	Mass of fiber 8–16 wt%, composting 40–45 °C, 90 days	Single pieces (10 mm long) were found at initial concentration of SPs 8%, and tangled fibers were noted at initial concentration of SPs 16%
Composite fibers from PLA, PLA/PHA, and PLA/PBS—13–26 μm [18]	Compost, pH 7.1, moisture 53%, 58 °C, 28 days	All fibers showed a decrease in strength and molecular weight
PCL—180.7 μm [96]	Composting (polymer 10 g, activated compost-144 g and sea sand—320 g), 58 °C, 56 days	Degradation—79.9%

* PHA—polyhydroxyalkanoates.

**Table 6 microorganisms-12-02024-t006:** The effect of HSs on biodegradation of SPs in anaerobic digestion.

SPs (Sample Size) [Reference]	MOs, Type of HSs, Conditions of AD	Result
PAM [99]	High-solid anaerobic digestion, dewatered sewage sludge (Anting Waste Water Treatment Plant) PAM—500 mg/L, pH 10.0, 120 rpm, 35 °C, 26 days	Removal of PAM—35.9%
PVA cryogel [100,101]	AD of oil desulfurization wastes, pH 7.0—8.5, 35 °C, 3 years. PVA cryogel is used as a carrier for the immobilization of MOs	There is no degradation of the SPs used as a cell carrier for 3 years under anaerobic conditions
PVA cryogel [98]	AD pH 7.0–8.5, 35 °C, 1–10 g/L HSs, 16 days. PVA cryogel is used as a carrier for the immobilization of MOs	

**AD**—anaerobic digestion.

**Table 7 microorganisms-12-02024-t007:** Characteristics of HSs involved in the biodegradation processes of SPs *.

Origin of HSs [Reference]	HA/FA	Ratio of Chemical Elements		
		H/C	C/N	O/C
**Biochar**				
Biochar * [17]	n/d ***	0.03–0.05	19.9–59.0	0.17–0.29
Biochar ** [106]	n/d	0.06	56.89	0.39
Biochar ** [107]	1.85–2.17	0.41–1.00	n/d	0.06–0.29
**Soil**				
HA (upper horizons of ferrallitic soil) * [87]	-	0.08	15.00	0.67
Compost soil * [108]	n/d	n/d	12.00	n/d
Agro-forestry soils ** [109]	5.20–12.30	1.1–1.8	11.60–19.90	n/d
Red clay soil (Utisol) ** [110]	0.90–2.50	0.92–2.07	15.56–40.98	1.14–4.63
**Compost**				
Chicken manure compost HA * [90]	-	n/d	16.6	0.27
Dairy manure compost HA * [90]	-	n/d	21.7	0.28
Sludge compost HA * [90]	-	n/d	31.4	0.24
Chicken manure/rice husk compost ** [111]	0.36–1.10	0.08–0.19	6.79–10.56	n/d
Municipal waste compost ** [112]	1.61–3.32	1.03–1.15	11.16–16.67	0.54–0.59
**Water sediments**				
lake sediments ** [113]	n/d	1.40–1.60	11.10–13.70	0.50–1.00
lake sediments ** [114]	n/d	0.12	8.10	0.62
river sediments ** [114]	n/d	0.08–0.11	8.7–14.0	0.69–0.78
river sediments ** [115]	0.39–3.32	1.00–2.20	9.43–43.48	0.48–1.38
lake sediments ** [116]	2.69–8.70	1.39–4.16	8.30–15.03	0.66–18.88
**Landfill/Anaerobic digestion**				
sewage sludge AD ** [117]	n/d	1.85	5.44	n/d
sewage sludge AD ** [118]	5.41	1.7–1.8	6.4–7.0	0.40–0.60

* An analysis of the characteristics of those HSs that participate in the processes from Table 1 is presented. ** The characteristics of HSs are taken from similar references [106,107,109,110,111,112,113,114,115,116,117,118] for comparative analysis. n/d—not detected. *** n/d—not detected.

**Table 8 microorganisms-12-02024-t008:** Destruction of SPs in the presence of HSs exhibiting photochemical activity.

SPs [Reference]	HSs	Degradation Conditions	Effects
PP-MPs [120]	HA	Laboratory xenon-lamp, temperature 60 °C, humidity 70%, light irradiance 1000 W/m^2^	The oxygen content of PP-MPs increased by 1.5% after the addition of HA.
PVC-MPs [121]	5 mg/L HA	100 mg/L PVC-MPs, laboratory photochemical reactor, 25 °C, 72 h	The presence of HA enhances the photoaging of PVC-MPs, while the presence of HA significantly increased the toxicity of the filtrate.
PS-MPs [15]	HA	Laboratory aqueous system, UV light (365 nm), irradiation intensity of 60 mW/cm^2^, 25 °C, PS-MPs:HA = 1:1 (*w*/*w*), 40 days	Weight loss of PS-MPs was increased by 4.3% in presence of HA. HA provided 42.62% increase in the concentration of oxygen-containing compounds within photodegradation of PS-MPs.
LDPE, PLA (thin films 4 × 2 cm with 150 µm thick) [122]	HA-doped/ ZnO nano- particles Molecular weight of HA-227.17 g/mol	Photocatalytic methods, based on the action of ROS. Laboratory aqueous system, ZnO-based photocatalytic material coated with LDPE and PLA films was placed in a bath containing water and exposed to UVA/light irradiation, UV range at 305 nm, 313 nm, and 366 nm and in the visible range at 404.7 nm and 435.8 nm, 25 °C 75 h and 225 h	HA/nanoparticles demonstrated improved photocatalytic activity for SPs compared to conventional ZnO.
PP-MPs (100–150 μm) [123]	HA (Suwannee River humic acid) and FA (Pony Lake fulvic acid) SRHA/PLFA	Photoaging, lake water in laboratory 0.5 g of PP MPs and 20 mL of 10 mg C/L (carbon content per liter of solution) SRHA/PLFA, 500 W mercury lamp, stirring at 700 rad/min, 25 °C	Photoaging of PP-MPs significantly slowed down in lake water compared to ultrapure water after 12 days of UV irradiation in the presence of HA and FA. The chromophores in HA and FA competed for photons with MPs through the light-shielding effect, thereby causing less fragmentation of PP-MPs.
PE [124]	HA granulate and FA powder were purchased from Omnia Specialities	Soil from different regions of Australia, natural conditions, 48 days	HA and FA increase rate of PE photo-oxidation. The impact of soil on PE photo-oxidation was complex and dependent at least in part on soil components that varied between different soil types, consequently influencing their photochemistry.

## Data Availability

Not applicable.

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
