# Peer review of "Role of Humic Substances in the (Bio)Degradation of Synthetic Polymers under Environmental Conditions"

_microorganisms, 2024, doi:10.3390/microorganisms12102024_

Round 1
Reviewer 1 Report
Comments and Suggestions for Authors
Dear authors,
The review presented is interesting and important to the literature. There are, however, a few subjects that must be adressed to improve the text:
1) The Table 1 is cited in the Figure 2 legend. Please, cite Table 1 in the text, not in the figure legend. I understand that they are related, however, this should be explained in the text.
2) In the abstract and introduction, mainly: the sentences are very long. Try to shorten them to make the text more fluidity. For example, there is a sentence in the Abstract that starts in the line 13 and goes on to the line 18.
3) Table 2: correct the second colunm in the last line and add the film size in the first colunm.
4) Table 2(second line, last colunm): "surface of"
5) Lines 316-321: sentence very long and confusing. I suggest you rewrite it.
6) Aryl group C-O? Alkyl ether C=O? Please confirm these information (line 427 and 428)
7) Table 6: first line - "ratio of"
8) Conclusion: sentences very long again, which make the text confusing.
Comments on the Quality of English Language
The quality of the text is fine.
Author Response
Dear Reviewer,
We are grateful to you for the suggestions allowing us the improving of our manuscript.
Please, see our comments to your remarks and the revised text of the paper:
According to your recommendations the following changes in the text were made:
Dear Reviewer,
We are grateful to you for the suggestions allowing us the improving of our manuscript.
Please, see our comments to your remarks and the revised text of the paper:
According to your recommendations the following changes in the text were made:
Reviewer #1:
Dear authors,
The review presented is interesting and important to the literature. There are, however, a few subjects that must be addressed to improve the text:
1) The Table 1 is cited in the Figure 2 legend. Please, cite Table 1 in the text, not in the figure legend. I understand that they are related, however, this should be explained in the text.
Response of the authors: According to recommendation of Reviewer the Title of Figure 2 was corrected. It was changed from:
“Figure 2. Principal scheme of theoretical analysis of biodegradation of synthetic polymer in the presence of HSs undertaken in this review. In Table 1 columns I, II contain information about biodegradation efficiency of SPs by various MOs and column III contains information about influence of HSs on MOs with corresponding references.”.
to:
“Figure 2. Principal scheme of theoretical analysis of biodegradation of synthetic polymer in the presence of HSs undertaken in this review: I, II - information about biodegradation efficiency of SPs by various MOs, III - information about influence of HSs on MOs with corresponding references [25-65]”.
2) In the abstract and introduction, mainly: the sentences are very long. Try to shorten them to make the text more fluidity. For example, there is a sentence in the Abstract that starts in the line 13 and goes on to the line 18.
Response of the authors: Lines 13-18, The Abstract was corrected and changed from:
“At this stage, there is a need to understand the relationship between the main participants in the processes of (bio)degradation of SPs in various ecosystems (reservoirs with fresh and sea water, soils, etc.), namely the polymers themselves, the cells of microorganisms (MOs) participating in their degradation, and humic substances (HSs), which constitute a macrocomponent of natural non-living organic matter of aquatic and soil ecosystems, formed and transformed in the processes of mineralization of bioorganic substances in environmental conditions.”
to:
“At this stage, there is a need to understand the relationship between the main participants in the processes of (bio)degradation of SPs in various ecosystems (reservoirs with fresh and sea water, soils, etc.), namely the polymers themselves, the cells of microorganisms (MOs) participating in their degradation, and humic substances (HSs). HSs constitute a macrocomponent of natural non-living organic matter of aquatic and soil ecosystems, formed and transformed in the processes of mineralization of bioorganic substances in environmental conditions.”.
On the recommendation of the reviewer, the phrases in the text of the article (not only in the Introduction) were shortened!
3) Table 2: correct the second column in the last line and add the film size in the first column.
Response of the authors: Table 2: Unfortunately, the correct film size cannot be added to the first column because it is absent in the referenced article. The text present in the second column in the last line was corrected.
4) Table 2 (second line, last column): "surface of"
Response of the authors: Table 2: The text in the second line (the last column) was corrected.
5) Lines 316-321: sentence very long and confusing. I suggest you rewrite it.
Response of the authors: Lines 316-321: the text was changed from:
” Thus, in vitro studies have confirmed the theoretically expected direct relationship between the presence and concentrations of HSs in media with SPs and the modification of the latter, improving their availability for biodegradation, enhancing microbial respiration, increasing the concentration of the biomass of active biofilms that carry out the biodegradation of SPs, including MPs, starting from the surface layers of polymeric materials.”.
to
” Thus, in vitro studies have confirmed the theoretically expected direct relationship between the presence and concentrations of HSs in media with SPs and the modification of the latter. Availability of HSs leads to improving their biodegradation, enhancing microbial respiration, increasing the concentration of the biomass of active biofilms that carry out the biodegradation of SPs, including MPs, starting from the surface layers of polymeric materials”.
6) Aryl group C-O? Alkyl ether C=O? Please confirm these information (line 427 and 428)
Response of the authors: Line 427 and 428: the text was corrected and changed from:
“(aryl C–O and alkyl ether =O)”
To “ (C(carbonyl)=O from alkyl esters and C(aryl)–O) ”
7) Table 6: first line - "ratio of"
Response of the authors: Text "ratio of" was corrected.
8) Conclusion: sentences very long again, which make the text confusing.
Response of the authors: The text in Conclusion was corrected.
With high respect and good wishes,
Authors of the manuscript.

Reviewer 2 Report
Comments and Suggestions for Authors
1.There are a lot of unnecessary abbreviations that make reading difficult as one has to remember all the time what abbreviation was what.I suggest that those abbreviations are only used for tables and figues and not in the text,
2. The manuscript is too "wordy", try to minimise the words and add more figures to make the manuscript more understandable,
3. The conclusion has no link to the main body of the manuscript and must have been placed in introduction or anywhere else but where it is now,
4. Figure 3 is the cream of the manuscript and needs to be fully described but in the manuscript, its explanation is too short,
5.In the conclusion, Isuggest to describe Figure 3 fully and with required detail.
Author Response
Dear Reviewer,
We are grateful to you for the suggestions allowing us the improving of our manuscript.
Please, see our comments to your remarks and the revised text of the paper:
According to your recommendations the following changes in the text were made:
Reviewer #2:
Comments and Suggestions for Authors
- There are a lot of unnecessary abbreviations that make reading difficult as one has to remember all the time what abbreviation was what. I suggest that those abbreviations are only used for tables and figures and not in the text,
Response of the authors: Taking into account the recommendation of the reviewer, the common abbreviations used for the plastics were disclosed in the article, the complete names of the polymers were given in the text, and abbreviations now present only in the tables and Figure 4.
- The manuscript is too "wordy", try to minimise the words and add more figures to make the manuscript more understandable,
Response of the authors: the revision of the text was done. Some phrases were shortened. The text was reduced in those places where it was possible. The two new pictures (figures 3 and 6) were added to the text of the article based on the data discussed. Since then, the numeration of the pictures was changed in the article.
- The conclusion has no link to the main body of the manuscript and must have been placed in introduction or anywhere else but where it is now.
Response of the authors: Conclusion was changed. Information performed at the end of the manuscript is based on the scientific analysis which was made by the authors of the article. Since then it cannot be placed in the Introduction. In the Introduction usually the well-known information can be performed. This reviewer is a “pioneer” work in the analysis of interactions disclosed between microorganisms, plastics, humic acids, water and soil conditions in the presence or absence of UV-rays of sun light. The most conclusions belong to the authors, they were accumulated step by step in this manuscript.
- Figure 3 is the cream of the manuscript and needs to be fully described but in the manuscript, its explanation is too short,
Response of the authors: Thank you for the recommendation. The text after the Figure 3 (now it is Figure 4) was added with the explanation of the picture and obtained results.
- In the conclusion, I suggest to describe Figure 3 fully and with required detail.
Response of the authors: Conclusion was changed. Partially it was replaced to the Introduction, and information coming from the results performed at Figure 3 was added to the text of Conclusion.
With high respect and good wishes,
Authors of the manuscript.

Round 2
Reviewer 2 Report
Comments and Suggestions for Authors
Thank you for doing the corrections.